# Gender Specificities in Sleep Disturbances following Mild Traumatic Brain Injury: A Preliminary Study

**DOI:** 10.3390/brainsci13020323

**Published:** 2023-02-14

**Authors:** Géraldine Martens, Mohammad Hossein Khosravi, Nicolas Lejeune, Jean-François Kaux, Aurore Thibaut

**Affiliations:** 1Coma Science Group, GIGA Consciousness, University of Liège, 4000 Liège, Belgium; 2Physical and Rehabilitation Medicine and Sport Traumatology Department, University Hospital of Liège, University of Liège, 4000 Liège, Belgium; 3Saint-Luc Hospital Group, CHN William Lennox, 1340 Ottignies, Belgium

**Keywords:** mild traumatic brain injury, concussion, sleep, gender

## Abstract

Mild traumatic brain injury (mTBI), or concussion, can lead to persistent cognitive and functional symptoms that impede quality of life to a varying extent. This condition is referred to as post-concussive syndrome (PCS). Sleep disturbances are part of it but their distribution among different genders remains scarcely investigated. This pilot cross-sectional anonymous web-based survey interviewed volunteer 18–55 years old participants with a recent (i.e., less than 5 years) reported history of mTBI. Questionnaires related to persistent post-concussive symptoms (PPCS; Rivermead post-concussion symptoms questionnaire), sleep quality (Pittsburgh Sleep Quality Index) and daytime sleepiness (Epworth Sleepiness Scale) were administered as part of the survey. Ninety-one participants’ responses were analysed (61 female; 30 male); 43% of them suffered from post-concussive syndrome, 60% reported poor sleep quality and 34% experienced excessive daytime sleepiness. The proportion of PPCS was significantly higher in female participants as compared to males (female: 57%; male: 13%; Fisher’s exact test *p* < 0.001). Excessive daytime sleepiness was also significantly more present in females (female: 44%; male: 13%; *p* < 0.001) whereas poor sleep quality was present in similar proportions between females and males (female: 66%; male: 50%; *p* = 0.176). Even though based on a relatively small sample, these findings highlight important gender differences that should be accounted for in PPCS medical care and management.

## 1. Introduction

Mild traumatic brain injury (mTBI), also referred to as concussion, occurs following the application of external mechanical or acceleration-deceleration forces to the brain. It induces brain micro-structures impairments leading to neural functional alterations with, typically, no objective structural lesions observed with conventional neuroimaging [1]. These functional alterations can be responsible for an array of cognitive, somatic, behavioural and sleep impairments [2]. These symptoms should resolve as the injury-related metabolites are cleared during the healing process in the days and weeks following the incident [3]. However, when the mTBI-related clinical symptoms remain and last over 4 weeks, they are considered as persistent post-concussive symptoms (PPCS), for which the condition is sometimes referred to as post-concussive syndrome [4]. This condition can have highly disabling consequences on the individuals’ quality of life and seems to affect females and males differently. Gender differences have previously been shown in athletic and general populations; the prevalence of mTBI is higher in females and they also report more PPCS and with a greater severity than males [5,6,7,8,9,10].

Among PPCS, sleep disturbances represent one of the unwanted consequences of mTBI. Half of the TBI population, regardless of the injury severity, suffer from sleep disturbances [11]. The impact of poor sleep on recovery from mTBI specifically is a critical factor [12]; indeed, sleep quality is a strong predictor for good recovery following mTBI in children and adults [13,14]. It is well known that the prevalence of sleep quality impairments and sleep disorders differs between genders [15,16]. The proportion of sleep disturbances is higher in females who reported a concussion [17]. However, in mTBI, the potential influence of gender on the combination of PPCS with sleep disturbances is still unclear.

The objective of this study was thus to investigate the relationship between sleep disturbances and PPCS in a study population representative of the general population while accounting for the potential impact of gender.

## 2. Materials and Methods

### 2.1. Study Procedures

The protocol of this cross-sectional study was reviewed by the central ethics committee of the University and University Hospital of Liege, which did not object to this anonymized survey (CE 2020-224). Following an introductory statement where participants consented to share their responses with the research team, this web-based survey was designed following three sections: (1) sociodemographic (age, gender, country, employment) and concussion-specific (number of mTBI/concussions, circumstances, diagnosing party) information; (2) assessment of PPCS wih the Rivermead post-concussion symptoms questionnaire (see below); and (3) assessment of sleep disturbances with the Pittsburgh Sleep Quality Index and the Epworth Sleepiness Scale (see below). It then underwent clinical experts’ consultation as well as translations so that the final version would be available in French, Dutch and English. For the recruitment, the survey was disseminated through a web-based invite with a link from January to March 2021 through institutional (i.e., University and University Hospital of Liège, GIGA Consciousness research lab) social media, professional associations of physicians and physiotherapists in Belgium and sports federations with a partnership with the University Hospital of Liège, Belgium. The inclusion criteria were as follows: age between 18 and 55 years old and diagnosed with a concussion by medical staff within the past 5 years. Any recipient of this web-based invite who considered meeting the inclusion criteria could participate on a voluntary basis.

### 2.2. Questionnaires

The Rivermead post-concussion symptoms questionnaire (RPQ) is a validated questionnaire that covers a set of 16 somatic, emotional and cognitive symptoms. Referring to the last 24 h, participants are asked to indicate the severity of each symptom on a 5-point Likert scale. The maximum total score is 64. A cut-off of ≥16 corresponds to clinical post-concussive symptomatology [18].

The Pittsburgh Sleep Quality Index (PSQI) is a self-reported questionnaire interrogating sleep quality over the past month using 19 items over 7 subcategories [19]. The composite score ranges from 0 (healthy sleep quality) to 21 (poor sleep quality). A cut-off of >5 points indicates a “poor sleeper”.

The Epworth Sleepiness Scale (ESS) is a self-reported questionnaire evaluating daytime sleepiness [20]. It encompasses 8 questions assessing the chance of dozing on a 4-point Likert scale. The total score ranges from 0 (normal) to 24 (excessive sleepiness) and a cut-off of ≥10 points indicates excessive daytime sleepiness.

### 2.3. Analyses

Descriptive statistics were provided using counts (n) and proportions (%) and variables estimates with means and standard deviations (SD). Differences between males and females were assessed using Chi-square tests and Fisher’s exact tests for dichotomous variables. Student’s *t*-tests were used for continuous variables. The relationship between PPCS (RPQ scores) and sleep disturbances (PSQI and ESS scores) was evaluated using Spearman’s correlation. Based on the cut-off scores provided above, participants were categorised following presence of PPCS (RPQ ≥ 16), poor sleep quality (PSQI > 5) and presence of daytime sleepiness (ESS ≥ 10). Results were considered significant at the *p* < 0.05 level.

## 3. Results

The outreach of the questionnaire could not formally be assessed as multiple diffusion sources were used (i.e., no return rate available). The questionnaire was filled in by 234 participants. The average completion time was 18 min. Among them, 80 were excluded based on the afore-mentioned inclusion criteria. Questionnaires that were partially filled (n = 63) were further excluded. The final study sample therefore comprised 91 participants; 61 identified as female (67%), 30 as male (33%) and none as other. Among them, 53 (58%) suffered from one concussion, 22 (24%) from two concussions and 16 (18%) from more than two. Sport-related concussions were the most prevalent (55%). The sociodemographic characteristics collected are presented in Table 1.

At the group level, the mean (±SD) RPQ score was 17.0 (±15.3); 39 (43%) participants scored at the ≥16 cut-off score, meaning they suffered from PPCS. Regarding sleep quality, the mean (±SD) PSQI score was 7.5 (±3.9); 55 (60%) participants scored at the >5 cut-off score and were therefore considered as “poor sleepers”. Finally, the mean (±SD) ESS score was 7.7 (±4.8); 31 (34%) participants scored at the ≥10 cut-off score and therefore experienced excessive daytime sleepiness. The questionnaires total scores are presented in Table 2.

### 3.1. Relationship between PPCS and Sleep Disturbances

There was a significant correlation between RPQ scores and PSQI scores (r = 0.54; *p* < 0.001). Likewise, there was a significant correlation between RPQ scores and ESS scores (r = 0.42; *p* < 0.001). Unsurprisingly, there was also a significant correlation between PSQI and ESS scores (r = 0.37; *p* < 0.001).

### 3.2. Gender Specificities

The number of mTBIs did not differ between male and female identifiers (*p* = 0.99). Regarding the presence of PPCS as assessed by the RPQ, there was a significant difference in the proportions of males (13%) vs. females (57%) who scored at the ≥16 cut-off (*p* < 0.001). For the poor sleep quality component assessed by the PSQI, there was no significant difference between males (50%) and females (66%) who scored at the >5 cut-off score (*p* = 0.176). Finally, for the presence of daytime sleepiness assessed by the ESS, there was a significant difference in the proportions of males (13%) vs. females (44%) who scored at the ≥10 cut-off score (*p* < 0.001). These results are presented in Table 2.

## 4. Discussion

This study aimed at evaluating the presence of PPCS and sleep disturbances following mTBI in a sample extracted from the general population and at comparing the gender proportions in participants reporting these troubles. We evaluated PPCS with the RPQ, which also encompasses an assessment of sleep but is not designed to evaluate sleep disturbances *per se*. We therefore used additional specific questionnaires (i.e., PSQI and ESS) to better capture these disturbances.

Our results show that a little less than a half of the study population suffer from PPCS. This is a concerning finding with regards to its impact on the quality of life but is in line with the current literature [21]. Sleep disturbances, as assessed by self-reported quality of sleep and daytime sleepiness, were also highly prevalent: the quality of sleep was insufficient for two thirds of the sample whereas daytime sleepiness was less reported (about a third). The significant relationship between PPCS and sleep disturbances confirm their interrelation and can guide clinicians caring for these troubles. Sleep management is indeed often neglected in the medical care following mTBI while it appears to be strongly related to the presence of persistent symptoms. Regarding the influence of gender, female participants suffered significantly more from PPCS following mTBI than their male counterparts. This was also the case for daytime sleepiness but not for sleep quality.

The amount of previous mTBIs sustained could have impacted the severity of PPCS [22,23] but in the present sample, this amount was similar for female and male participants. It thus appears that females are more prone to PPCS and sleepiness following mTBI as compared to males and, based on the available data collected in this survey, this pattern does not seem to be influenced by the amount of mTBIs reported.

Several explaining factors can be suggested based on current hypotheses in the mTBI literature. The higher amount and severity of PPCS in females has already been previously reported [5,6,7] and is partly attributed to sociocultural factors. Indeed, females tend to admit easier having symptoms (e.g., fatigue, dizziness, troubles concentrating) whereas some cultural and/or societal norms on masculinity can make it more difficult more males to report them [6]. This is especially true in the sports context were males are more expected to be tough and to neglect pain than females [24]. This difference in reporting patterns leads to a reporting bias and might apply to the sleep-related outcomes as well, captured here with the PSQI and ESS questionnaires that rely on self-reported symptoms. Crossing these reports with objective clinical data (e.g., polysomnography) in female and male PPCS patients reporting sleep disturbances would be of interest for future research.

In addition, the hormonal differences might also affect the reported symptoms, including sleep disturbances. The neuroprotective versus detrimental effect of estrogen and the primary female hormone, is debated [24] and requires additional human studies. Hormonal birth control has also been associated with greater symptom burden [25] and the menstrual phase in which the injury occurs further appears to have an impact on mTBI outcomes [26,27]. This suggests a hormonal influence on mTBI symptomatology but also warrants dedicated human studies.

### Limitations

This study encompasses several limitations preventing the generalizability of the results. First, for an online survey, the sample size is low and affects the study power and the analyses. The results would need to be confirmed on a larger sample in follow-up studies. Second, the population is not well balanced in terms of gender representation and thereby does not accurately reflect the general population of patients with mTBI. The overrepresentation of females in our sample might be caused by the voluntary nature of the study and the differences in symptom reporting between females and males mentioned earlier. Even though suboptimal, this sample size already showed significant gender differences. Third, potential confounders affecting the outcomes (e.g., medication, comorbidities) have not been collected in order to keep the survey short and optimize the completion rate. Finally, our sample did not include participants identifying as other than cis- male or female. Future studies should better target these communities in order to address this current research gap.

## 5. Conclusions

This pilot data on a volunteer sample of mTBI patients show the important interrelations between persisting symptoms (PPCS) and disturbed sleep. The severity of PPCS is indeed related to the severity of sleep disturbances. Regarding the influence of gender, female participants do not only report greater severity of PPCS, but they also report experiencing significantly more excessive daytime sleepiness as compared to males. These specificities should be considered and accounted for in daily clinical care.

## Figures and Tables

**Table 1 brainsci-13-00323-t001:** Sociodemographic characteristics and concussion history of the study sample (n = 91) according to gender.

Variable	Female (n = 61) N (%)	Male (n = 30) N (%)	χ^2^; *p* Value
**Sociodemographic**			
Age group (years)			4.80; 0.19
18–24	18 (29)	13 (44)	
25–34	23 (38)	7 (23)	
35–44	8 (13)	7 (23)	
45–55	12 (20)	3 (10)	
Country			
Belgium	45 (74)	24 (80)	
France	3 (5)	4 (13)	
Luxembourg	2 (3)	0 (0)	
Netherlands	11 (18)	0 (0)	
Other	0 (0)	2 (7)	
Employment			
Employee	24 (39)	8 (27)	
Executive	3 (5)	2 (7)	
Student	15 (25)	14 (46)	
Unemployed	1 (2)	0 (0)	
Worker	4 (6)	3 (10)	
Other	14 (23)	3 (10)	
**mTBI history**			
Number of mTBIs			0.86; 0.93
One	36 (59)	17 (57)	
Two	15 (25)	7 (23)	
Three	7 (11)	3 (10)	
Four	1 (2)	1 (3)	
More than four	2 (3)	2 (7)	
Circumstance of last mTBI			4.14; 0.39
Car accident	10 (16)	3 (10)	
Domestic accident	8 (13)	2 (7)	
Sport	29 (48)	21 (70)	
Work-related	3 (5)	1 (3)	
Other	11 (18)	3 (10)	
First diagnosing party			
Emergency/hospital	38 (62)	21 (70)	
General practitioner	14 (23)	2 (7)	
Specialist physician	3 (5)	4 (13)	
Sports coach	2 (3)	1 (3)	
Other	4 (7)	2 (7)	

**Table 2 brainsci-13-00323-t002:** Total scores for the Rivermead Post-concussion symptoms questionnaire (RPQ), the Pittsburgh Sleep Quality Index (PSQI) and the Epworth Sleepiness Scale (ESS) for the whole sample and by gender.

Questionnaire	All (n = 91)	Female (n = 61)	Male (n = 30)	χ^2^; *p* Value ^1^
**Rivermead Post-concussion symptoms questionnaire (RPQ)**				
RPQ total score (/64); m (sd)	17.0 (±15.3)	21.6 (±14.7)	7.7 (±11.9)	*p* < 0.001
RPQ above cut-off ^2^; n (%)	39 (43)	35 (57)	4 (13)	*p* < 0.001
**Pittsburgh Sleep Quality Index (PSQI)**				
PSQI total score (/21); m (sd)	7.5 (±3.9)	7.8 (±3.8)	6.8 (±4.1)	*p* = 0.240
PSQI above cut-off ^3^; n (%)	55 (60)	40 (66)	15 (50)	*p* = 0.176
**Epworth Sleepiness Scale (ESS)**				
ESS total score (/24); m (sd)	7.7 (±4.8)	8.7 (±5.2)	5.6 (±3.3)	*p* = 0.004
ESS above cut-off ^4^; n (%)	31 (34)	20 (44)	4 (13)	*p* < 0.001

^1^ Student’s t-test for continuous variables, Fisher’s exact test for categorical variables. ^2^ RPQ above cut-off score ≥16 indicating PPCS. ^3^ PSQI above cut-off score >5 indicating poor sleepers. ^4^ ESS above cut-off score ≥10 indicating excessive daytime sleepiness.

## Data Availability

Data related to this study will be made available upon reasonable request to the corresponding author (geraldine.martens@uliege.be).

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
