# Peer review of "Gender Specificities in Sleep Disturbances following Mild Traumatic Brain Injury: A Preliminary Study"

_brainsci, 2023, doi:10.3390/brainsci13020323_

Round 1

Reviewer 1 Report

brainsci-2104556: Gender specificities in sleep disturbances following mild traumatic brain injury.

In this study, the authors are trying to demonstrate the sleep differences in male and female patients suffered from concussion(s).

     However, this attempt is evidently unsuccessful because of: a) very small database, b) unequal gender populations, and c) omitted age-dependent gender distributions.

Author Response

Response to Reviewer 1 Comments

Point 1: In this study, the authors are trying to demonstrate the sleep differences in male and female patients suffered from concussion(s).

However, this attempt is evidently unsuccessful because of: a) very small database

Response 1: We acknowledge our database is small. This was a pilot study conducted to help us design follow-up studies in our institutions. We also faced an important amount of incomplete questionnaires. We still believe this is important data as it specifically assesses persistent post-concussional symptoms (PPCS) in relation with sleep disturbances, as opposed to most studies focusing on mild traumatic brain injury (mTBI), gender and sleep without assessing PPCS with a validated questionnaire such as the Rivermead post-concussion symptoms questionnaire (RPQ). To avoid misleading our readers, we now acknowledge the small nature of our database in our abstract and our discussion, where we added a dedicated section on limitations.

Point 2: b) unequal gender populations

Response 2: Indeed, the gender distribution is unequal which reflects, in our view, the reporting bias favoring female self-reported outcomes versus male ones. This is in itself represent relevant data since it confirms this reporting bias. We now discuss this further in the limitations section of the discussion.

Point 3: c) omitted age-dependent gender distributions

Response 3: We now report and analyse gender distribution across age. Our statistical test do not show any significant difference between age categories and gender.  

Reviewer 2 Report

The authors present a short report on a small sample of participants after completing several surveys. These data illustrate greater post-concussive symptoms in female vs male participants, including sleep specific outcomes (PSQI and ESS). I have a few suggestions and comments.

First, the debate regarding sex differences in the prevalence of these symptoms may be ongoing, but nevertheless has been extensively reported on. A quick pubmed search to refamiliarize myself with this recent literature was a nice reminder. Increase PCS in females vs males is not necessarily novel information. Similarly, the reported sex effects on sleep are also not particularly novel. 

Apart from the potential lack of novelty, there are concerns with the study design. Inclusion criteria was based on age and TBI status, nothing else. There are potential issues related to reporting bias, i.e.., are women more likely to respond than men; and related to comorbidities (highly numerous, e.g., sleep apnea, chronic pain, depression, medication usage, etc.). 

To make such conclusions with the employed approach, a much larger sample would be needed to draw meaningful statistical comparisons. Regarding the analysis, please also report mean scores across groups and whether or not those comparisons reached statistical significance. If still significant, what happens if you adjust for age in this comparison? 

Nevertheless, this remains an ongoing topic of interest. I would highly encourage the authors to review more thoroughly the literature in this space and include a more comprehensive assessment of the overall landscape regarding the effect of sex on PCS and sleep impairment following mTBI. This would specifically address shortcomings in the introduction and discussion. The goal being to provide a more balanced presentation of the field.

Finally, this manuscript needs a limitations section describing thoroughly the many caveats therein.

An optional suggestion would also include revising figures to be produced in a more graphically friendly format, 

Author Response

Response to Reviewer 2 Comments

Point 1: The authors present a short report on a small sample of participants after completing several surveys. These data illustrate greater post-concussive symptoms in female vs male participants, including sleep specific outcomes (PSQI and ESS). I have a few suggestions and comments.

First, the debate regarding sex differences in the prevalence of these symptoms may be ongoing, but nevertheless has been extensively reported on. A quick pubmed search to refamiliarize myself with this recent literature was a nice reminder. Increase PCS in females vs males is not necessarily novel information. Similarly, the reported sex effects on sleep are also not particularly novel.

Response 1: We acknowledge the novelty aspect is limited. We agree there is available research on PCS in females vs males and on the reported sex effects on sleep. However the combination of PCS, gender/sex and sleep is less common. The current litterature also tend to focus on sport-related concussion only while the present study focused on the general population.

Point 2: Apart from the potential lack of novelty, there are concerns with the study design. Inclusion criteria was based on age and TBI status, nothing else. There are potential issues related to reporting bias, i.e.., are women more likely to respond than men; and related to comorbidities (highly numerous, e.g., sleep apnea, chronic pain, depression, medication usage, etc.).

Response 2: We acknowledge the inclusion criteria were not very restrictive. Such criteria are challenging to control for in an anonymous web-based survey and therefore we wanted to have as few criteria as possible. In addition, we believe that our large inclusion criteria allow to collect data of a representative sample of the general population. The reporting bias is an intereseting point we now extend on in the discussion section. Regarding comorbidities, we agree they would have a significant impact on our collected outcomes. However, we wanted to limit the amount of question in our survey in order to maximize the completion rate. We now acknowledge this limitation in our discussion section.

Point 3: To make such conclusions with the employed approach, a much larger sample would be needed to draw meaningful statistical comparisons. Regarding the analysis, please also report mean scores across groups and whether or not those comparisons reached statistical significance. If still significant, what happens if you adjust for age in this comparison?

Response 3: We agree a larger sample would be needed and therefore emphasize the pilot aspect of this study which will guide our follow-up studies. Regarding the analysis, we now also report mean scores across groups and compare them. It reached statistical significance for the RPQ and ESS scores but we could not adjust for age given the categorical value of the variable. We could however demonstrate that the age distributions were similar between males and females.

Point 4: Nevertheless, this remains an ongoing topic of interest. I would highly encourage the authors to review more thoroughly the literature in this space and include a more comprehensive assessment of the overall landscape regarding the effect of sex on PCS and sleep impairment following mTBI. This would specifically address shortcomings in the introduction and discussion. The goal being to provide a more balanced presentation of the field.

Response 4: We thank the reviewer for the relevant feedback. We now extent the current litterature on the topic in the introduction and discussion sections.

Point 5: Finally, this manuscript needs a limitations section describing thoroughly the many caveats therein.

Response 5: We followed the reviewer’s suggestion and added a limitations section describing the caveats related to sample size, gender representation and additional confounders.

Point 6: An optional suggestion would also include revising figures to be produced in a more graphically friendly format

Response 6: As part of our responses to all the reviewers’ concerns, we replaced the figure with a table to provide more clarity on the total scores across all groups.

Reviewer 3 Report

The authors have assessed the impact of concussion on sleep patterns, comparing men to women, via a self-administered survey. They find that post-concussive symptoms were more common in women than men. Excessive daytime sleepiness was more common in women, but poor sleep quality was similar between men and women.  These findings are interesting. No mechanism to explain these differences has been identified.

Several changes could improve the manuscript, as follows:

1. The METHODS do not clearly state how patients were identified to be requested to complete the survey.

2. The DISCUSSION should be expanding to suggest possible mechanisms to explain the results.

3. The manuscript does not include a specific CONCLUSION of three to four sentences for closure. 

Author Response

Response to Reviewer 3 Comments

Point 1: The authors have assessed the impact of concussion on sleep patterns, comparing men to women, via a self-administered survey. They find that post-concussive symptoms were more common in women than men. Excessive daytime sleepiness was more common in women, but poor sleep quality was similar between men and women. These findings are interesting. No mechanism to explain these differences has been identified.

Several changes could improve the manuscript, as follows:

  1. The METHODS do not clearly state how patients were identified to be requested to complete the survey.

Response 1: We thank the reviewer for this feedback. We now explicit in the methods sections (study procedures) how the participants were recruited.

Point 2: The DISCUSSION should be expanding to suggest possible mechanisms to explain the results.

Response 2: We followed the reviewer’s suggestions and now expand on possible mechanisms (i.e., hormonal, sociocultural) explaining the results in the discussion section.

Point 3: The manuscript does not include a specific CONCLUSION of three to four sentences for closure.

Response 3: We now added a specific conclusion section at the end of our manuscript.

Reviewer 4 Report

Thank you for submitting this paper. Although a very interesting topic I believe that your results require a much greater discussion. You are maintaining that from your survey it would seem that females are more likely to suffer sleep disturbance than males, however to fail to discuss the important element of reporting bias. This is an extremely important point since it may significantly impact on your results.

Even though you are publishing a "brief report" I believe that withough the discussion of these limitations incorrect assumptions can easily be made. 

Detailed Comments:

1. Abstract: This cross-sectional anonymous web-based survey 14
interrogated
volunteer 18-55 years old participants with a recent (i.e., less than 5 years) reported 15
history of mTBI.

Comments: "interrogated"  is a very strong term in English - would you not prefer a term such as interviewed?

2. Introduction

Likewise, the prevalence and severity of mTBI is reportedly higher in females as compared to males, especially in athletic populations.

Comments: Conclusion from the Dick article After evaluating multiple years of concussion data in comparable sports, the evidence indicates that female athletes may be at greater risk for concussion than their male counterparts. There also is some evidence that gender differences exist in outcomes of traumatic brain injury and concussions. Because concussion is a clinical diagnosis often depending on self reporting and with no established biological marker or consistent symptoms/definitions, and because there is evidence that females are more honest in reporting general injuries than males, it is unclear whether the concussion incidence data, while generally consistent in showing a higher risk in females as compared to males in similar sports, is a true difference or is influenced by a reporting bias.

This statement conflicts with much of what is reported regarding head injuries. I have read the references you are citing but Dick himself acknowledges that there may well be reporting bias hence the large number of females.

3. Discussion

(1) The amount of previous mTBIs sustained could have impacted the severity of PPCS (REF) but in the present sample, this amount was similar for female and male participants.

Comments: Are you intending to add a reference here?

(2) This discussion is somewhat lacking in content. Much more is required here. Indeed it is important to documents the limitations of your study and moreover there are many references indicating that females are much more likely to answer surveys in general and to self-report concussion specifically.

4. References

Ref.7-Incomplete reference

Ref.8-Incomplete reference

Ref.15-Incorrect referencing style

Author Response

Response to Reviewer 4 Comments

Point 1: Thank you for submitting this paper. Although a very interesting topic I believe that your results require a much greater discussion. You are maintaining that from your survey it would seem that females are more likely to suffer sleep disturbance than males, however to fail to discuss the important element of reporting bias. This is an extremely important point since it may significantly impact on your results. Even though you are publishing a "brief report" I believe that without the discussion of these limitations incorrect assumptions can easily be made.

Response 1: We thank the reviewer for this feedback. The reporting bias is indeed an important aspect we now extend on in the discussion section. We further acknowledge in our limitations section how it could have impacted our results.

Point 2: Detailed Comments:

  1. Abstract: This cross-sectional anonymous web-based survey interrogated volunteer 18-55 years old participants with a recent (i.e., less than 5 years) reported history of mTBI.

Comments: "interrogated" is a very strong term in English - would you not prefer a term such as interviewed?

Response 2: We thank the reviewer and followed the suggestion of using “interviewed” instead.

Point 3: Introduction

Likewise, the prevalence and severity of mTBI is reportedly higher in females as compared to males, especially in athletic populations.

Comments: Conclusion from the Dick article After evaluating multiple years of concussion data in comparable sports, the evidence indicates that female athletes may be at greater risk for concussion than their male counterparts. There also is some evidence that gender differences exist in outcomes of traumatic brain injury and concussions. Because concussion is a clinical diagnosis often depending on self reporting and with no established biological marker or consistent symptoms/definitions, and because there is evidence that females are more honest in reporting general injuries than males, it is unclear whether the concussion incidence data, while generally consistent in showing a higher risk in females as compared to males in similar sports, is a true difference or is influenced by a reporting bias.

This statement conflicts with much of what is reported regarding head injuries. I have read the references you are citing but Dick himself acknowledges that there may well be reporting bias hence the large number of females.

Response 3: We fully agree with the reviewer’s point. Our study was not designed to assess the actual incidence of concussion between male and females since we fully relied on self-reported outcomes and did not include additional objective data such a biomarkers.

We now balance this view by discussing the gender reporting bias and how it could affect both mTBI and sleep-related outcomes in our discussion section.

Point 4: Discussion

(1) The amount of previous mTBIs sustained could have impacted the severity of PPCS (REF) but in the present sample, this amount was similar for female and male participants.

Comments: Are you intending to add a reference here?

Response 4: Yes this was an omission. We now added the references we intended to place there.

Point 5: Discussion

(2) This discussion is somewhat lacking in content. Much more is required here. Indeed it is important to documents the limitations of your study and moreover there are many references indicating that females are much more likely to answer surveys in general and to self-report concussion specifically.

Response 5: We thank the reviewer for the feedbacks and suggestions. We now significantly extended our discussion section by adding aspects related to reporting bias and hormonal factors as well as adding limitations and conclusion sections.

Point 6: References

Ref.7-Incomplete reference

Ref.8-Incomplete reference

Ref.15-Incorrect referencing style

Response 6: Thank you for pointing this out. We corrected the incomplete and incorrect referecences in our updated version of the manuscript.

Round 2

Reviewer 1 Report

brainsci-2104556-v2: “Gender specificities in sleep disturbances following mild trau-2 matic brain injury”.

The authors have made a careful revision and reasonably responded to almost all points I raised. However, the “ambitious” title should be clarified by the adding of “a pilot study” or ”a preliminary study”.

Author Response

Point 1: The authors have made a careful revision and reasonably responded to almost all points I raised. However, the “ambitious” title should be clarified by the adding of “a pilot study” or ”a preliminary study”.

Response 1: We thank the reviewer for the feedback and followed the suggestion of clarifying the title. It now reads “Gender specificities in sleep disturbances following mild traumatic brain injury: a preliminary study”.

Reviewer 2 Report

Thank you, I have no further comments.

Author Response

Point 1: Thank you, I have no further comments.

Response 1: We thank the reviewer for the time and feedback.

Reviewer 3 Report

The authors have addressed the reviewer's comments. The recruitment of potential survey candidates is still a little unclear - my interpretation of the authors' description is that the e-mail was sent to the University of Liege community ("institutional" is not clearly defined). If this interpretation is incorrect, rewording may be helpful. Otherwise, the comments are satisfactorily addressed.

Author Response

Point 1: The authors have addressed the reviewer's comments. The recruitment of potential survey candidates is still a little unclear - my interpretation of the authors' description is that the e-mail was sent to the University of Liege community ("institutional" is not clearly defined). If this interpretation is incorrect, rewording may be helpful. Otherwise, the comments are satisfactorily addressed.

Response 1: We thank the reviewer for this feedback. To clarify the recruitement aspect, we now specify the institutions. It now reads: “For the recruitment, the survey was disseminated through a web-based invite with a link from January to March 2021 through institutional (i.e., University and University Hospital of Liège, GIGA Consciousness research lab) social media, professional associ-ations of physicians and physiotherapists in Belgium and sports federations with a partnership with the University Hospital of Liège, Belgium.”